# Epidemiology of *Mycobacterium tuberculosis* lineages and strain clustering within urban and peri-urban settings in Ethiopia

Hawult Taye[1,2]*, Kassahun Alemu[2], Adane Mihret[1], Sosina Ayalew[1], Elena Hailu[1], James L. N. Wood[3], Ziv Shkedy[2,4], Stefan Berg[5], Abraham Aseffa[1], The ETHICOBOTS consortium[¶]

1 Armauer Hansen Research Institute, Addis Ababa, Ethiopia, 2 Department of Epidemiology and Biostatistics, Institute of Public Health, College of Medicine and Health Sciences, University of Gondar, Gondar, Ethiopia, 3 Disease Dynamics Unit, Department of Veterinary Medicine, University of Cambridge, Cambridge, United Kingdom, 4 Biostatistics and Bioinformatics, University of Hasselt, Hasselt, Belgium, 5 Bacteriology Department, Animal and Plant Health Agency, New Haw, United Kingdom

¶ Members of the ETHICOBOTS consortium are listed under Acknowledgments.
* hawultachew@gmail.com

**Data Availability Statement:** All relevant data are within the paper and its Supporting Information files.

## Abstract

### Background

Previous work has shown differential predominance of certain *Mycobacterium tuberculosis* (*M. tb*) lineages and sub-lineages among different human populations in diverse geographic regions of Ethiopia. Nevertheless, how strain diversity is evolving under the ongoing rapid socio-economic and environmental changes is poorly understood. The present study investigated factors associated with *M. tb* lineage predominance and rate of strain clustering within urban and peri-urban settings in Ethiopia.

### Methods

Pulmonary Tuberculosis (PTB) and Cervical tuberculous lymphadenitis (TBLN) patients who visited selected health facilities were recruited in the years of 2016 and 2017. A total of 258 *M. tb* isolates identified from 163 sputa and 95 fine-needle aspirates (FNA) were characterized by spoligotyping and compared with international *M.tb* spoligotyping patterns registered at the SITVIT2 databases. The molecular data were linked with clinical and demographic data of the patients for further statistical analysis.

### Results

From a total of 258 *M. tb* isolates, 84 distinct spoligotype patterns that included 58 known Shared International Type (SIT) patterns and 26 new or orphan patterns were identified. The majority of strains belonged to two major *M. tb* lineages, L3 (35.7%) and L4 (61.6%). The observed high percentage of isolates with shared patterns (n = 200/258) suggested a substantial rate of overall clustering (77.5%). After adjusting for the effect of geographical variations, clustering rate was significantly lower among individuals co-infected with HIV and other concomitant chronic disease. Compared to L4, the adjusted odds ratio and 95%

**Funding:** This work was funded by the Biotechnology and Biologic Sciences Research Council, the Department for International Development, the Economic & Social Research Council, the Medical Research Council, the Natural Environment Research Council and the Defence Science & Technology Laboratory, under the Zoonoses and Emerging Livestock Systems (ZELS) program, ref: BB/L018977/1. SB was also partly funded by the Department for Environment, Food & Rural Affairs, United Kingdom, ref: TBSE3294. The Armauer Hansen Research Institute is supported by core funds from Norad and Sida.

**Competing interests:** The authors have declared that no competing interests exist.

confidence interval (AOR; 95% CI) indicated that infections with L3 *M. tb* strains were more likely to be associated with TBLN [3.47 (1.45, 8.29)] and TB-HIV co-infection [2.84 (1.61, 5.55)].

## Conclusion

Despite the observed difference in strain diversity and geographical distribution of *M. tb* lineages, compared to earlier studies in Ethiopia, the overall rate of strain clustering suggests higher transmission and warrant more detailed investigations into the molecular epidemiology of TB and related factors.

## Introduction

Tuberculosis (TB) is a chronic infectious disease caused by species of the *Mycobacterium tuberculosis* complex (MTBC). Except for *Mycobacterium tuberculosis (M. tb)*, which is the primary cause of human TB, other members of the MTBC are believed to have adapted to different animal hosts and therefore they may have reduced fitness to cause human infection [1, 2]. Beside environmental and socio-economic factors, the biology and epidemiology of human TB has likely been shaped by the historical interaction between MTBC members and its host [2, 3]. The genetic variation between MTBC species contributes to the ambiguities concerning disease presentation, frequency of transmission and clinical progress [2, 4]. This is particularly true for *M. tb*, where the interaction of genotypic variation among different strains with human genetic polymorphism play a prominent role in the epidemiology of TB diseases [4–7]. The overall epidemiology of MTBC species is influenced by the environment, with its frequency and distribution being dependent on social, economic, and ecological causes [4, 8]. Although, there are no well-established classical factors that are known to be strongly associated with disease phenotype, immunological studies have suggested that some *M. tb* strains and lineages are more virulent and/or more infectious than others [9]. It has been stated that some strains that belong to the modern MTBC Lineages are more capable of inducing higher inflammatory response than lineages of the same clade (Haarlem, high; Beijing, low) [10]. However, difference in pathogenicity and lineage specific rate of transmission are important only when considered together with the host genotype and geographical location [11].

Although, it is still challenging to investigate the influence of bacterial and host genotype on the development of different forms of TB in humans, disease phenotype seems to be associated with a bacterial genotype [2, 6]. According to other published reports, L4 seemed more likely to be associated with Pulmonary TB (PTB) while L2 and L3 were linked with extra-pulmonary TB (EPTB) disease, such as TB meningitis and TB in cervical Lymph Nodes (TBLN) [12–15]. Another comparative study showed that strains of the East African Indian (L3) and Euro-American (L4) lineages were negatively associated with extra thoracic disease as compared to strains of the East Asian lineage (L2) [16]. These studies thereby suggest that species diversity and their interaction with host biology affects the pathophysiology and natural course of TB disease [2, 17]. For example, a study conducted in Tanzania has shown that chronic signs of TB disease, such as weight loss, have been more associated with L4 strains than with strains of the Indo-Oceanic (L1) lineage [18]. In addition to factors associated with human genetics such as ethnicity, biological and clinical determinants of an individual, such as HIV and body mass index, have shown significant difference on disease phenotype and rate of transmission across major *M. tb* Lineages [16, 19–21].

Different alternative molecular identification methods have been used to estimate rates of disease transmission, which is generally inferred by comparing genotypic clustering between patient isolates from a given epidemiological setting [10, 22]. In other words, successful transmission of particular genotypes has been reflected through an increase in the frequency and consistency of strain domination over time in defined populations [16, 23]. However, despite recently developed advanced molecular diagnostic tools, both the nature of genotype variations and the characteristics of the host immune response to certain types of *M. tb* strains are largely unknown in many TB high burden settings [24, 25]. Particularly in countries like Ethiopia, where there is high prevalence and high transmission rate and a diversified population of bacterial species [26–29], molecular identification of the agents can be an important component of the knowledge base required to improve on previous achievements of the national TB control program. Taking all this into account, the present study investigated factors associated with *M. tb* lineage predominance and rate of strain clustering within the context of urban and peri-urban settings in Ethiopia.

## Materials and methods

### Study design and setting

A multi-centre health facility based cross-sectional study was conducted in Ethiopia during 2016 and 2017. As part of the Ethiopia Control of Bovine Tuberculosis Strategies (ETHICO-BOTS) project, four hospitals, two private clinics, and fourteen health centers located in urban and peri-urban areas, were purposively selected from four different regions of Ethiopia. Addis Ababa was the largest study site and constituted of Addis Ababa city and the surrounding special zone of Oromiya region while the remaining three study sites were located in the regional urban cities of Mekele in Tigray, Gondar in Amhara, and Hawassa in Southern Nations Nationalities, and Peoples' region.

### Study population

Recruitment of participants at selected health facilities was carried out according to the national guideline standard case definition criteria. All presumed TB cases were initially considered as potential source of the study population. Then those patients clinically diagnosed with PTB or TBLN were asked for informed consent and enrolled consecutively. Recruitment of PTB cases was done at all selected governmental health facilities. TBLN patients were enrolled from all four study sites; however they were only recruited from the Pathology Units of three governmental hospitals and two private clinics because of lack of diagnostic facilities and skilled professionals for fine-needle aspirate (FNA) cytology examination at governmental health centers. Included cases from both groups were those eligible for first-line Anti-TB treatment. Known MDR (multi drug resistant) TB cases and EPTB patients other than those with TBLN were excluded in this study.

### Data collection

Clinical and demographic information was collected from recruited TB cases using a pre-tested structured questionnaire. Following the routine care service, consented PTB and TBLN participants were requested to provide spot sputum and FNA samples, respectively. Care providers (nurses) working at directly observed therapy (DOT) centres collected sputum specimens using sterile containers. FNA specimens were collected from the selected hospitals and private clinics by experienced pathologists who performed FNA cytology examination as part of their routine diagnostic service. According to the standard procedure, FNA collection was

performed using a 21-gauge needle attached to a 10 ml syringe and specimens were collected into cryo-tubes with sterile phosphate buffer saline (PBS). Samples were kept at -20˚C at remote study sites until transported on ice boxes to the Armauer Hansen Research Institute (AHRI) TB laboratory where the clinical samples were stored at -80˚C until processed for mycobacterial culture. Clinical sample handling and laboratory procedures were performed according to a previously published protocol [27].

## Mycobacterial culturing

Samples collected in the study were processed and cultured for mycobacteria using standard procedures established at the AHRI TB laboratory [27, 30]. Specimen samples were inoculated on Löwenstein-Jensen (LJ) medium slants supplemented with either glycerol or pyruvate and incubated at 37˚C. The slopes were examined weekly for up to eight weeks for any visible growth. Bacterial colonies identified as Acid-Fast Bacilli by ZN staining [27] were saved as frozen stocks in 20% glycerol as well as heat-inactivated in 500μl distilled $H_2O$ at 80˚C for 60 min; the latter samples were used for subsequent molecular identification.

## Molecular identification techniques

All isolates were screened by Large Sequence Polymorphism (LSP) typing using conventional PCR for examination of Region of Difference 9 (RD9) according to protocols by Berg et al. (2009) [31]. Spoligotyping was performed according to Kamerbeek et al. (1997) [32], using a non-commercial biodyne-C-membrane produced by the Animal & Plant Health Agency (United Kingdom).

## Genotype analysis and comparison with global databases

Spoligotype patterns were converted into binary and octal formats and compared with previously reported strains in the international SITVIT2 database [8] hosted by Institute Pasteur de la Guadeloupe. Here, spoligotypes shared by more than one strain were designated as shared types and were assigned a shared international type (SIT) number according to the SITVIT2 database, while patterns that were not recognized in the latest online version of the database were labelled as "New" if the pattern was identified for more than one strain and "Orphan" if the pattern was unique to only one strain. Further lineage classification for corresponding nomenclature was done using the 'Run TB-Lineage' online tool from linked databases (http://www.miru-vntrplus.org/MIRU/index.faces and http://tbinsight.cs.rpi.edu/run_tb_lineage.html). Here, major lineages were predicted using a conformal Bayesian network (CBN) analysis while knowledge based Bayesian network (KBBN) analysis was used to predict the corresponding sub-lineages.

## Data management and statistical analysis

All genotype outputs from the computer assisted analyses were imported to SPSS and merged with clinical and demographic data. The final clean dataset was exported to STATA and R-software to perform further statistical analysis. Two of the main outcome variables, clustering rate and *M. tb* lineages, were categorized as binomial scale of measurement. In the first category, "clustered" referred to two or more isolates sharing identical spoligotyping patterns while isolates that did not have shared patterns was defined as "unique". Here, three different logistic regression analysis methods were performed to identify and compare factors associated with strain clustering. The first Bivariable analysis was performed to estimate a crude (unadjusted) odd ratio for each independent categorical variable while the second multivariable

logistic regression analysis was used to estimate adjusted odd ratio (AOR with 95% CI) that better reflect the likelihood of included variable associated with rate of strain clustering. The third model (hierarchical logistic regression) was preferred to adjust for the effect of regional variations, the first level factor that often attributed with strain clustering, where host-related clinical factors and spoligotype-based *M. tb* lineage classification were considered as second level factors. Variables included in the second model were reconsidered and used to compare the corresponding adjusted estimates (AOR with 95% CI) generated from the third (Multi-level) model which was done using STATA software with the recommended (melogit) command. The multivariable logistic regression was used to determine the clinical characteristics or disease phenotypes associated with dominant *M. tb* lineage. In both cases, R-package Soft-ware commands were used to perform bivariable and multivariable logistic regression. Before running the multivariable logistic regression analysis, stepwise backward elimination tech-nique was applied to select independent variables. Initially, all clinically relevant factor vari-ables were included in the full model. Then using the specific statistical command (Step) under R-studio, the software program automatically generated all possible alternative models having lists of dependent and independent variables. Finally, according to the Likelihood Ratio-test and to minimize the effect of confounding variables, a relatively better fitted model with potential explanatory variables that has the lowest akaki information criteria (AIC) was selected. Independent relationship of variables was decided based on different cut-off point for statistical significance level ($\alpha$: $< 0.05$; $< 0.01$ and $< 0.001$) and interpretation of key findings was reported using the adjusted estimates (AOR with 95% CI).

### Ethical considerations

This study was part of the ETHICOBOTS project, which obtained ethical clearance from the Federal Ministry of Science and Technology (Ref. No: 301/001/2015), the AHRI/ALERT Ethics Review Committee (Project Reg. No: PO46/14) and from University of Gondar Institutional Review Board (Review number: O/V/P/RCS/04/45/2016). Support letters were obtained from Regional State Health Bureaus and health facilities. Enrollment of study participants was done after written informed consent was secured and signed agreements were received from all par-ticipating health facilities. Detailed information about the risks and benefits of the study as well as confidentiality of the research data was a prerequisite for study participation.

## Results

### Characteristics of the study population

This study examined a total of 258 TB patients (163 PTB and 95 TBLN cases) of which 145 (56.2%) were male and 113 (43.8%) were female, with a mean age of 32.2 (±12.9) years. Most of these TB cases were from Gondar, 111/258 (43.0%), and Mekele, 61/258 (23.6%), in north-ern Ethiopia while the remaining patients, 44/258 (17.1%) and 42/258 (16.3%), were from Addis Ababa and Hawassa in central and southern Ethiopia, respectively. Farmers (80/258, 31.0%) and students (40/258, 15.5%) were the two most common occupations in the study population. With regard to the medical history of the participants, 20/258 (7.8%) were co-infected with HIV and 96/258 (37.2%) had at least one additional chronic concomitant disease (Table 1).

### Genetic diversity of *Mycobacterium tuberculosis* lineages

All 258 isolates provided in the S1 Table were genotyped by LSP as *M. tb* while being intact for RD9. When the isolates were spoligotyped 84 different patterns were identified, of which 58

**Table 1. Characteristics of the 258 study participants, 163 patients with pulmonary TB and 95 with cervical TB lymphadenitis, recruited at selected health facilities located in urban and peri-urban areas of Ethiopia in the years 2016/17.**

| Patient characteristics | PTB n (%) | TBLN n (%) | Total n (%) | P-value of Chi-square test |
|---|---|---|---|---|
| Number of patients | 163 (63.2%) | 95 (37%) | 258 (100%) | - |
| Age group | | | | |
| < 35 years | 105 (64.4) | 61 (64.2) | 166 (64.3) | 0.298 |
| ≥ 35 years | 58 (35.6) | 34 (35.8) | 92 (35.7) | |
| Gender | | | | |
| Male | 107 (65.6) | 38 (40.0) | 145 (56.2) | 0.000 |
| Female | 56 (34.4) | 57 (60.0) | 113 (43.8) | |
| Occupation | | | | |
| Farmer | 46 (28.2) | 34 (35.8) | 80 (31.0) | 0.087 |
| Merchant | 14 (8.6) | 11 (11.6) | 25 (9.7) | |
| Employee | 24 (14.7) | 9 (9.5) | 33 (12.8) | |
| Student | 24 (14.7) | 16 (16.8) | 40 (15.5) | |
| House wife | 20 (12.3) | 17 (17.9) | 37 (14.3) | |
| Dairy worker | 12 (7.4) | 4 (4.2) | 16 (6.2) | |
| Others | 23 (14.1) | 4 (4.2) | 27 (10.5) | |
| Geographical location | | | | |
| Gondar | 84 (51.5) | 27 (28.4) | 111 (43.0) | 0.000 |
| Hawassa | 34 (20.9) | 8 (8.4) | 42 (16.3) | |
| Mekele | 40 (24.5) | 21 (22.1) | 61 (23.6) | |
| Addis Ababa | 5 (3.1) | 39 (41.1) | 44 (17.1) | |
| HIV co-infection | | | | |
| No | 145 (89) | 93 (97.9) | 238 (92.3) | 0.010 |
| Yes | 18 (11) | 2 (2.1) | 20 (7.8) | |
| Chronic concomitant disease | | | | |
| No | 98 (60.1) | 64 (67.4) | 162 (62.8) | 0.246 |
| Yes | 65 (39.9) | 31 (32.6) | 96 (37.2) | |

SIT patterns were already recognized in the SITVIT2 database (accounting for 231/258 (89.5%) of the isolates). Among these patterns, 32 *M. tb* isolates were singletons while 25 designated shared patterns, each with 2 to 40 isolates, accounted for 85.7% (198/231) of all isolates with identified SIT patterns. The remaining twenty five unique orphan patterns and two isolates with a new shared spoligotype pattern (Table 2), representing 27 (10.5%) of the total isolates, were not yet recognized by the SITVIT2 database. As presented in Table 3, over half of the isolates 145/258 (56.2%) were represented by five of the dominant SIT patterns, including SIT25 (n = 40), SIT149 (n = 36), SIT53 (n = 32), SIT26 (n = 17), and SIT37 (n = 11).

According to the CBN analysis, 97.3% of the total 258 isolates belonged to two major lineages, EA (61.6%) and EAI (35.7%). On the basis of SNP-based genome-wide phylogeny analysis, these lineages are commonly known as L4 and L3, respectively [2]. The remaining 7/258 (2.7%) were represented by IO (L1) and AFRI (L7), each with three strains, and one with the typical Beijing (L2) spoligotype pattern (Fig 1; S1 Table).

The alternative KBBN classification showed a predominance of the CAS (34.9%) sub-lineage among strains defined as L3. T (15.9%), T3-ETH (15.1%) and Haarlem (10.9%) were the most common sub-lineages of L4. There was a significant difference in geographical distribution between strain types; all LAM families of L4 (LAM, LAM3 and LAM5) were observed in the northern part of the country (Gondar and Mekele). Similarly, the CAS families (L3), which were highly dominant in the Gondar area, were rather rare around Hawassa. The Manu,

**Table 2. Descriptions of all orphan and new spoligotype patterns (n = 26) that were identified from 27 clinical samples collected from pulmonary TB and cervical TB lymphadenitis patients recruited at selected health facilities in Ethiopia in the years of 2016/17.**

| No | Spoligotype patterns of orphan or new strains | | Lineage classification based on | | | # of isolates |
|---|---|---|---|---|---|---|
| | Octal code | Binary format (presence (black) or absence (white) of 43 spacers) | KBBN | CBN | SNP-based prediction* | |
| 1 | 000001777020771 | | T1-RUS2 | EA | L4 | 1 |
| 2 | 037677560020771 | | H1 | EA | L4 | 1 |
| 3 | 101774000000000 | | ZERO | EA | L4 | 1 |
| 4 | 403000377760771 | | T1-RUS2 | EA | L4 | 1 |
| 5 | 477777757000771 | | H4-Ural-2 | EA | L4 | 1 |
| 6 | 503777740003171 | | CAS1-Delhi | EAI | L3 | 1 |
| 7 | 511777400003171 | | CAS | EAI | L3 | 1 |
| 8 | 555777437740171 | | T | EA | L4 | 1 |
| 9 | 603777700003771 | | CAS1-Delhi | EAI | L3 | 1 |
| 10 | 676777660760771 | | T | EA | L4 | 1 |
| 11 | 703737740003571 | | CAS1-Delhi | EAI | L3 | 1 |
| 12 | 703777700001171 | | CAS1-Delhi | EAI | L3 | 2 |
| 13 | 703777740001171 | | CAS1-Delhi | EAI | L3 | 1 |
| 14 | 703777740003171 | | CAS1-Delhi | EAI | L3 | 1 |
| 15 | 703777740003771 | | CAS1-Delhi | EAI | L3 | 1 |
| 16 | 703777747776771 | | Manu1 | EA | L4 | 1 |
| 17 | 711777740003171 | | CAS1-Delhi | EAI | L3 | 1 |
| 18 | 773777776000771 | | H3-Ural-1 | EA | L4 | 1 |
| 19 | 776737737760771 | | T3 | EA | L4 | 1 |
| 20 | 777000277760771 | | T3-ETH | EA | L4 | 1 |
| 21 | 777001777760771 | | T3-ETH | EA | L4 | 1 |
| 22 | 777737401760771 | | LAM5 | EA | L4 | 1 |
| 23 | 777737777760000 | | X2 | EA | L4 | 1 |
| 24 | 777777401760771 | | LAM | EA | L4 | 1 |
| 25 | 777777777420571 | | H3-Ural-1 | EA | L4 | 1 |
| 26 | 777777777600631 | | H3 | EA | L4 | 1 |

KBBN: knowledge based Bayesian network; CBN: conformal Bayesian network; SIT: shared international type; EA: Euro-American; EAI: East-African-Indian; IO: Indio-Oceanic.

* Supported by SNP typing (Firdessa et al 2013)

Haarlem and T families (all of L4) accounted for the majority of strains identified in the Hawassa region (Fig 2).

## Factors associated with strain clustering and predominance

The overall clustering rate aggregated from 26 (25 SIT and one new) shared patterns was 77.5% (200/258). Our multivariable analysis (Table 4) showed that as compared to Gondar, rate of clustering in Mekele and Hawassa was more than two and three fold higher, with adjusted OR (95% CI) of 2.71 (1.16, 6.34) and 3.56 (1.09, 11.63), respectively. However, an increased rate of *M. tb* transmission is generally inferred by comparing clustered genotyping patterns of clinical isolates from a given epidemiological setting [10]. By contrast, cases with

**Table 3. Spoligotype descriptions of all registered SIT patterns with two or more isolates identified from 198 clinical samples collected from pulmonary TB and cervical TB lymphadenitis patients recruited at selected health facilities in Ethiopia in the years of 2016/17.**

| Spoligotype patterns of shared SIT strains | | | Lineage classification | | | Shared isolates |
|---|---|---|---|---|---|---|
| SIT Nº | Octal code | Binary format (presence (black) or absence (white) of 43 spacers) | KBBN | CBN | SNP-based Prediction* | |
| 4 | 000000007760771 | | T1-RUS2 | EA | L4 | 2 (0.8) |
| 952 | 603777740003771 | | CAS1-Delhi | EAI | L3 | 3 (1.2) |
| 1729 | 700000004177771 | | AFRI | AFRI | L7 | 2 (0.8) |
| 21 | 703377400001771 | | CAS1-Kili | EAI | L3 | 5 (1.9) |
| 2359 | 703677740003171 | | CAS1-Delhi | EAI | L3 | 4 (1.6) |
| 2973 | 703701740003171 | | CAS1-Delhi | EAI | L3 | 2 (0.8) |
| 1199 | 703701740003171 | | CAS1-Delhi | EAI | L3 | 2 (0.8) |
| 25 | 703777740003171 | | CAS1-Delhi | EAI | L3 | 40 (15.5) |
| 26 | 703777740003771 | | CAS1-Delhi | EAI | L3 | 17 (6.6) |
| 1877 | 737377777760771 | | T | EA | L4 | 2 (0.8) |
| 33 | 776177607760771 | | LAM3 | EA | L4 | 3 (1.2) |
| 149 | 777000377760771 | | T3-ETH | EA | L4 | 36 (14.0) |
| 504 | 777737737760771 | | T3 | EA | L4 | 2 (0.8) |
| 726 | 777737747413771 | | EAI6-BGD1 | IO | L1 | 2 (0.8) |
| 35 | 777737777420771 | | H3-Ural-1 | EA | L4 | 2 (0.8) |
| 37 | 777737777760771 | | T3 | EA | L4 | 11 (4.3) |
| 1688 | 777777403760771 | | LAM | EA | L4 | 2 (0.8) |
| 41 | 777777404760771 | | Turkey | EA | L4 | 5 (1.9) |
| 121 | 777777775720771 | | H3 | EA | L4 | 4 (1.6) |
| 817 | 777777777420731 | | H3-Ural-1 | EA | L4 | 2 (0.8) |
| 777 | 777777777420771 | | H3-Ural-1 | EA | L4 | 2 (0.8) |
| 134 | 777777777720631 | | H3 | EA | L4 | 2 (0.8) |
| 52 | 777777777760731 | | T2 | EA | L4 | 5 (1.9) |
| 53 | 777777777760771 | | T | EA | L4 | 32 (12.4) |
| 54 | 777777777763771 | | Manu2 | EA | L4 | 9 (3.5) |

KBBN: knowledge based Bayesian network; CBN: conformal Bayesian network; SIT: shared international type; EA: Euro-American; EAI: East-African-Indian; IO: Indio-Oceanic.

* Supported by SNP typing (Firdessa et al 2013)

isolates of a unique pattern could be considered to have resulted from reactivation of latent infection or were else presumably acquired outside of the study population [33]. Considering that hierarchical logistic regression analysis was performed to minimize the observed heterogeneity due to geographical location. After controlling for the effect of regional variations adjusted estimates generated from the final model showed that the rate of strain clustering was inversely associated with TB-HIV co-infection and comorbidity with other chronic illnesses. As shown in Table 4, TB-HIV co-infected individuals [0.16 (0.05, 0.47)] and those who had any other concomitant chronic disease [0.46 (0.23, 0.91)] were less likely to have clustered strains as compared to patients diagnosed with only TB disease.

A second multivariable analysis was performed in relation to the clinical characteristics of the two most predominant lineages (L3 and L4). As shown in Table 5, in comparison to L4 strains of *M. tuberculosis*, the odds for TBLN cases infected with L3 was three and half fold

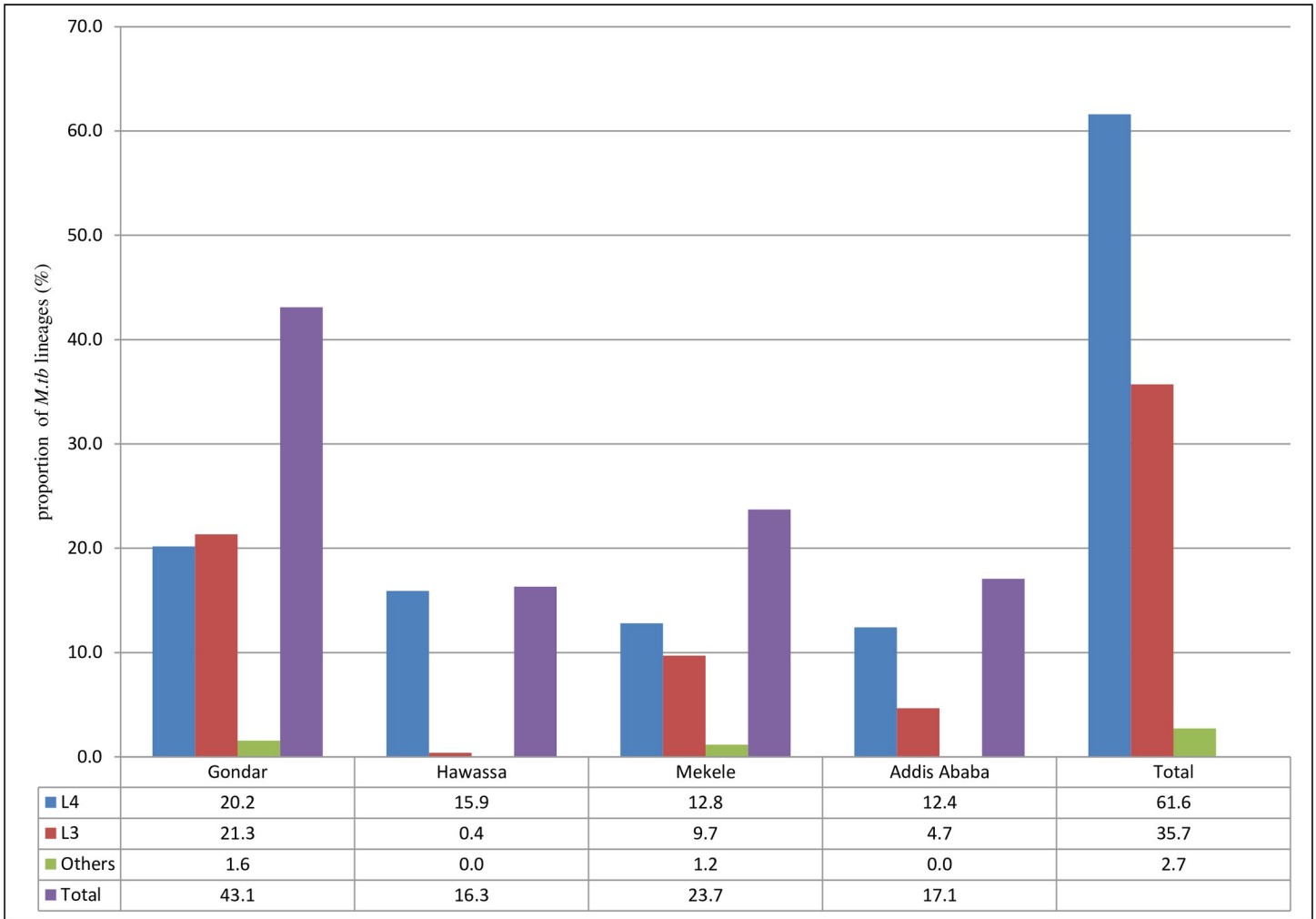

**Fig 1. Proportion of major *Mycobacterium tuberculosis* lineages circulating within peri-urban and urban areas in Ethiopia.** 'Others' include L7 (AFRI), L2 (Beijing), and L1 (IO).

[3.47 (1.45, 8.29)] higher than PTB patients. Active TB disease due to L3 strains was significantly associated with HIV-TB co-infection [2.84 (1.61, 5.55)], but less likely to be associated with concomitant chronic disease [0.46 (0.25, 0.87)], as compared to L4.

## Discussion

Despite the observed difference in strain diversity and distribution of *M. tb* lineages across regions, high percentage of shared patterns suggested a substantial overall strain clustering rate around urban and peri-urban settings in Ethiopia. Altogether, a predominance of known SIT patterns resulted in an overall strain clustering rate of 77.5% in the current study, with a range of 69–88% across the study regions (Table 4). That was significantly higher as compared to earlier Ethiopian studies (2005–2018) reviewed by Mekonnen et al. (2019), with a pooled clustering rate (95% CI) of 0.41 (0.32–0.50) [34]. Understandably, at national level, some population groups have likely contributed more to such TB incidence rate than other groups. Particularly, the risk of TB transmission around urban areas is known to be higher than among sparsely populated societies and rural communities [24, 29]. Because of the simultaneously

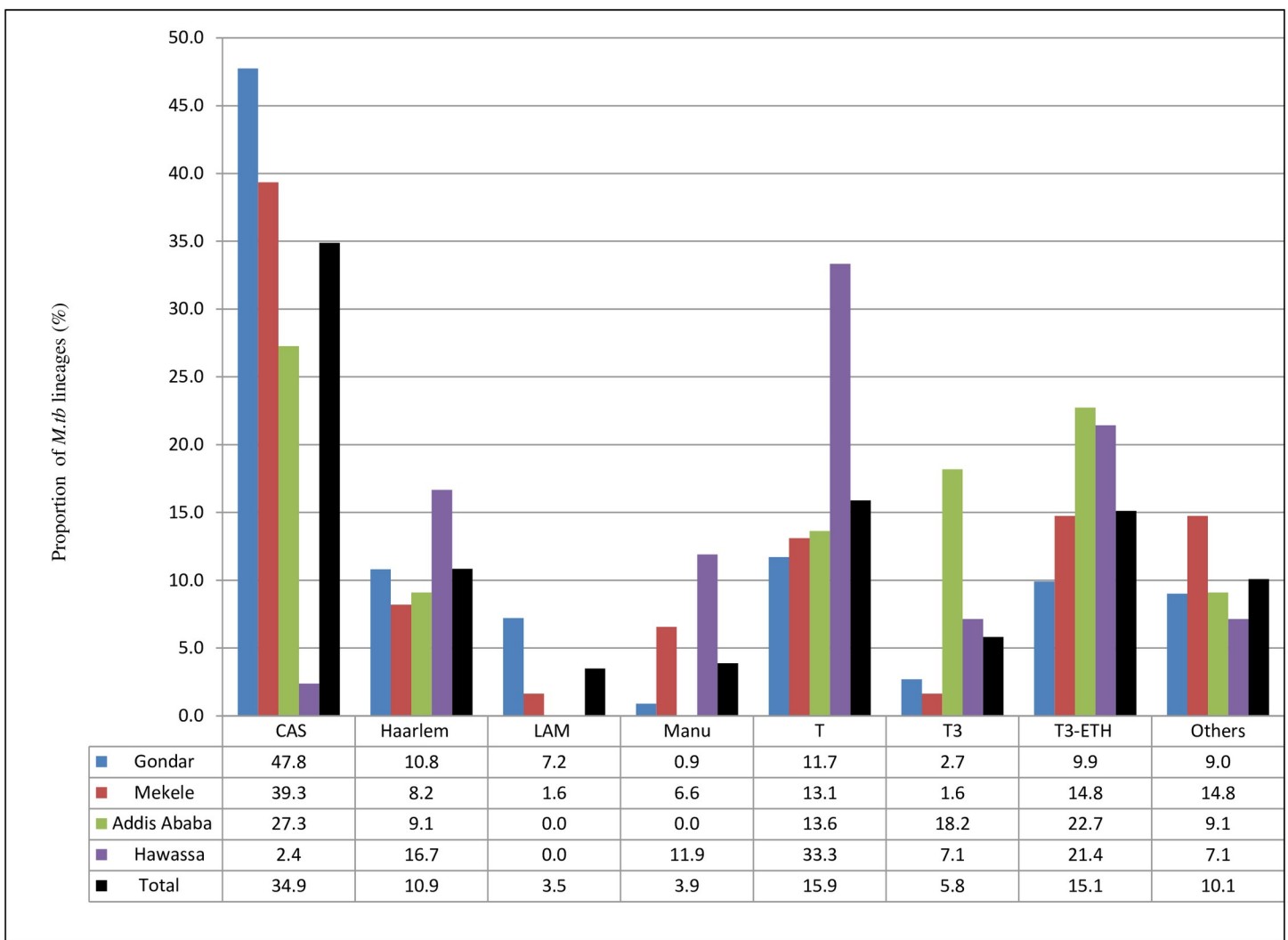

| | CAS | Haarlem | LAM | Manu | T | T3 | T3-ETH | Others |
|---|---|---|---|---|---|---|---|---|
| ■ Gondar | 47.8 | 10.8 | 7.2 | 0.9 | 11.7 | 2.7 | 9.9 | 9.0 |
| ■ Mekele | 39.3 | 8.2 | 1.6 | 6.6 | 13.1 | 1.6 | 14.8 | 14.8 |
| ■ Addis Ababa | 27.3 | 9.1 | 0.0 | 0.0 | 13.6 | 18.2 | 22.7 | 9.1 |
| ■ Hawassa | 2.4 | 16.7 | 0.0 | 11.9 | 33.3 | 7.1 | 21.4 | 7.1 |
| ■ Total | 34.9 | 10.9 | 3.5 | 3.9 | 15.9 | 5.8 | 15.1 | 10.1 |

**Fig 2. KBBN based classification *of Mycobacterium tuberculosis* sub-lineages circulating within peri-urban and urban areas in Ethiopia.** H1, H3, H3-Ural-1 and H4-Ural-2 were classified as 'Haarlem'; 'LAM' include LAM3 and LAM5; Manu represent Manu1 and Manu2. 'Others' include the following types: T2, Turkey, T1-RUS2, AFRI (Ethiopian), Beijing, EAI4-VNM, and EAI6-BGD1.

ongoing expansion of urbanization and emerging socio-economic conditions around urban areas in Ethiopia (increasing population size and density e.g. through expanding slums, congregation into condominiums, growing manufacturing and service sector), the pattern of TB transmission among those living and working in the urban and peri-urban areas is postulated to differ in strain diversity and clustering, compared to that of the general population [29], the majority (85%) of which are rural communities. Despite previous achievements in reducing national TB morbidity and mortality [35], summarized reports of data from the global burden of TB diseases in the last two decades have shown a declined rate in reducing the prevalence and mortality ratio in Ethiopia. Essentially, there has been a higher rate of new TB cases (incidence) in the last few years than what was expected from the previous trend [35, 36].

Accordingly, a diverse range of strains of *M. tb* lineages, many previously not registered in spoligotyping databases, continue to circulate and maintain a high rate of transmission of TB in Ethiopia. Similarly, as would be expected, the observed diversified type of *M. tb* strain and

**Table 4. Conventional and hierarchical (multi-level) logistic regression modeling methods were used to identify factors associated with strain clustering based on spoligotyping.**

| Factor variables | Proportion of cases n (%) | | Three logistic regression analyses | | |
| --- | --- | --- | --- | --- | --- |
| | | | Bivariable | Multivariable | Hierarchical |
| | Clustered | Unique | COR (95% CI) | AOR (95% CI) | AOR (95% CI) |
| Region | | | | | |
| Gondar | 77 (38.3) | 34 (59.6) | Ref | Ref | Level-I factor |
| Hawassa | 37 (18.4) | 5 (8.8) | 3.17 (1.14, 8.79)* | 3.56 (1.09, 11.63)* | |
| Mekele | 51 (25.4) | 10 (17.5) | 2.19 (0.99, 4.82) | 2.71 (1.16, 6.34)* | |
| Addis Ababa | 36 (17.9) | 8 (14.0) | 1.93 (0.81, 4.59) | 2.42 (0.84, 7.01) | |
| Diagnosis | | | | | |
| PTB | 127 (63.2) | 36 (63.2) | Ref | Ref | Ref |
| TBLN | 74 (36.8) | 21 (36.8) | 0.97 (0.53, 1.79) | 0.52 (0.24, 1.15) | 0.58 (0.27, 1.23) |
| HIV co-infection | | | | | |
| No | 191 (95.0) | 47 (82.5) | Ref | Ref | Ref |
| Yes | 10 (5.0) | 10 (17.5) | 0.27 (0.11, 0.71)** | 0.16 (0.05, 0.50)** | 0.16 (0.05, 0.47)*** |
| Co-morbidity of Chronic illness | | | | | |
| No | 134 (66.7) | 28 (49.1) | Ref | Ref | Ref |
| Yes | 67 (33.3) | 29 (50.9) | 0.50 (0.27, 0.91)* | 0.50 (0.25, 1.01) | 0.46 (0.23, 0.91)* |
| Hemoptysis | | | | | |
| No | 167 (83.1) | 42 (75.0) | Ref | Ref | Ref |
| Yes | 34 (16.9) | 14 (25.0) | 0.61 (0.30, 1.24) | 0.50 (0.22, 1.16) | 0.55 (0.24, 1.25) |
| TB lineage | | | | | |
| L3 (EAI) | 76 (37.8) | 16 (28.1) | Ref | Ref | Ref |
| L4 (EA) | 121 (60.2) | 38 (66.7) | 0.69 (0.36, 1.32) | 0.42 (0.20, 0.90)* | 0.49 (0.23, 1.04) |
| Others | 4 (2.0) | 3 (5.3) | 0.28 (0.06, 1.38) | 0.25 (0.04, 1.48) | 0.25 (0.04, 1.44) |

EA, Euro-American; EAI, East Africa-India; The cut-off point for statistical significance ($\alpha$) is represented by: $< 0.05 = $*; $< 0.01 = $**; $< 0.001 = $***

lineage distribution in the current study closely matched with studies analyzed in the two most recent TB reviews that showed specific lineage predominance across different geographical locations in Ethiopia [29, 34]. This means, the same two major lineages, L4 and L3 (Fig 1), were predominant [29, 30, 34], as were the five most common SIT patterns (Table 2) [14, 29, 37, 38]. As shown in Figs 1 and 2, the observed significant difference in proportions of strain types across the four study sites, has also been noted from previous studies in Ethiopia [29, 34]. Those less prevalent *M. tb* lineages, which included the Ethiopian (L7), the Beijing (L2), and the IO (L1) lineages, were identified from samples collected at sites located in the northern regions (Gondar and Mekele). Strains of L7, which was first reported by Firdessa et al [14, 28, 37, 39] and that seem highly confined to Ethiopia, remain more prevalent in the north of the country. The two SIT patterns (SIT1729 and SIT910) that we identified in this region are the same as for those strains that were previously classified as L7 [8, 14].

Taking into account the observed geographical difference, the current study investigated the contribution of bacterial genotype and host related factors associated with rate of strain clustering. While comparing clustered genotyping patterns of the two most predominant *M. tb* lineages, a relatively higher percentage of shared L3 patterns were identified as compared to clustered patterns that belonged to L4. Despite limited discriminatory power of the spoligotyping method, an increased rate of *M. tb* transmission is generally inferred by comparing clustered genotyping patterns of clinical isolates from a given epidemiological setting [10]. In contrast, cases with isolates of a unique pattern could be considered to have resulted from

**Table 5. Results of logistic regression analysis exploring associations between clinical characteristics and active TB disease caused by L3 versus L4, the two most dominant *Mycobacterium tuberculosis* lineages identified in the study.**

| Clinical characteristics | Proportion of Cases: n (%) | | Bivariable analysis | | Multivariable analysis | |
|---|---|---|---|---|---|---|
| | Lineage 3 | Lineage 4 | COR (95% CI) | P-value | AOR (95% CI) | P-value |
| Region | | | | | | |
| Addis Ababa | 12 (13.0) | 32 (20.1) | Ref | | Ref | |
| Gondar | 54 (58.7) | 53 (33.3) | 2.77 (1.29, 5.95) | 0.009 | 5.24 (2.03, 13.51) | < 0.001 |
| Hawassa | 1 (1.1) | 41 (25.8) | 0.07 (0.01, 0.53) | 0.010 | 0.11 (0.01, 0.95) | 0.044 |
| Mekele | 25 (27.2) | 33 (20.8) | 2.02 (0.87, 4.69) | 0.102 | 4.28 (1.52, 11.99) | 0.006 |
| Gender | | | | | | |
| Male | 56 (60.9) | 88 (55.3) | Ref | | Ref | |
| Female | 36 (39.1) | 71 (44.7) | 0.79 (0.47, 1.33) | 0.371 | 0.91 (0.48, 1.72) | 0.781 |
| Diagnosis | | | | | | |
| PTB | 53 (57.6) | 107 (67.3) | Ref | | Ref | |
| TBLN | 39 (42.4) | 52 (32.7) | 1.5 (0.88, 2.55) | 0.134 | 3.47 (1.45, 8.29) | 0.005 |
| HIV co-infection | | | | | | |
| No | 81 (88.0) | 151 (95.0) | Ref | | Ref | |
| Yes | 11(12.0) | 8 (5.0) | 2.93 (1.09, 7.85) | 0.033 | 2.84 (1.61, 5.55) | 0.027 |
| Comorbidity of Chronic illness | | | | | | |
| No | 62 (67.4) | 95 (59.7) | Ref | | Ref | |
| Yes | 30 (32.6) | 64 (40.3) | 0.73 (0.43, 1.25) | 0.252 | 0.46 (0.25, 0.87) | 0.016 |
| Taking prescribed Medication | | | | | | |
| No | 55 (59.8) | 117 (73.6) | Ref | | Ref | |
| Yes | 37 (40.2) | 42 (26.4) | 1.86 (1.08, 3.21) | 0.026 | 1.67 (0.83, 3.36) | 0.152 |
| Persistent Cough | | | | | | |
| No | 19 (20.7) | 32 (20.1) | Ref | | Ref | |
| Yes | 73 (79.3) | 127 (79.9) | 0.94 (0.49, 1.78) | 0.844 | 1.03 (0.41, 2.61) | 0.944 |
| Hemoptysis | | | | | | |
| No | 74 (80.4) | 129 (81.6) | Ref | | Ref | |
| Yes | 18 (19.6) | 29 (18.4) | 1.08 (0.56, 2.08) | 0.813 | 2.10 (0.90, 4.87) | 0.085 |
| Weight loss | | | | | | |
| No | 12 (13.0) | 27 (17.0) | Ref | | Ref | |
| Yes | 80 (87.0) | 132 (83.0) | 1.37 (0.66, 2.86) | 0.397 | 1.00 (0.41, 2.47) | 0.997 |

reactivation of latent infection or were else presumably acquired from outside of the study population [2, 33, 40]. Indeed, diverse *M. tb* strains could be identified in the different regions [2, 5, 8]. In spite of the fact that the molecular epidemiology of TB has shown remarkable difference across geographical locations, risk of transmission and TB disease progression is likely to depend on the interactions of various factors related to strain type and host immunity [8]. Bacterial genetic difference has been shown to have an impact on the extent of TB transmission; thus strains from TB lineages referred to as 'modern' lineages (L2-L4) are assumed to be more transmissible than other MTBC strains [2, 34]. It is interesting to note that after adjusting for the effect of regional variations, the likelihood of clustering was significantly lower among HIV co-infected patients and those who had any other concomitant chronic diseases. A higher risk of primary exposure or an increased rate of TB transmission in endemic settings has often been associated with the presence of more infectious PTB cases [41]. On the other hand, poor host immunity has been linked with endogenous reactivation of latent infection and could have greater contribution to the development of TBLN or disseminated TB [38]. However, as previously reported by others in several studies [14, 34, 37, 41], we also did not

observe any difference in clustering rate with respect to site of infection. This might be because of limited power of the study that could not control for all possible effects of confounding factors. Although, the differences in strain virulence and immunogenicity have been investigated in experimental studies, whether this phenotypic variation plays a role in human disease remains unclear [3, 6].

Therefore, it is believed that investigating the clinical epidemiology of dominant *M. tb* lineages among host populations would allow understanding of possible host-pathogen interaction. In this regard, one of the findings that emerged from this study is that clinical factors, which are often associated with host immunity, appeared to differ significantly between L3 and L4, the two most dominant lineages. According to the multivariate analysis (Table 5), the likelihood of detecting L3 among TBLN cases and HIV co-infected patients was significantly higher than for L4. However, a summary report generated from the updated version of the international *Mycobacterium tuberculosis* spoligotyping global database has shown a higher rate of CAS (L3) infection among HIV co-infected cases than other widely prevalent sub-lineages [8]. The observed discrepancy might be due to the interaction effect of sub-lineages or the possibility of co-infection within the same host. Our analysis was performed based on major *M. tb* lineage classification. Although it is often associated with host immunity, Osório et al. (2018) stated that due to selective advantage of extrinsic factors, within-host bacterial diversity seems to contribute to difference in disease progression [4]. For example, certain groups of L4 strains are found to be more virulent in terms of disease severity and to display higher rates of human-to-human transmission, but only at some specific geographical locations [2]. In favour of that, and as compared to L4, the current study identified significantly lower rate of L3 strains among TB cases diagnosed with other concomitant chronic illnesses (Table 5). Certainly, any immune-compromised condition and HIV interferes with bacterial virulence might lead to endogenous reactivation [20, 25, 41], suggesting that less virulent MTBC species could progress to active TB disease in immune-compromised patients. For example, TB patients infected with *M. africanum* were more likely to be older, HIV infected, and severely malnourished than those infected with *M. tb* [42]. Although the mechanisms are not yet clear, the influence of bacterial and host genotype on the development of different forms of TB in humans is well documented. In this regard, the findings observed in this study seem to agree with others that suggested a possible relationship between L3 and EPTB disease [12, 38]. Correspondingly, a significantly higher rate of PTB was often associated with L4, while more EPTB disease, such as TB meningitis and TBLN, was attributed to L3 [13, 15, 38].

Generally, because of a complex network related with many other proximal and distal determinants, *M. tb* strain clustering or lineage specific effects on disease presentations may not always be fully explained by some particular risk factors and it is difficult to quantify the biological effect using numerical estimates [43]. As a result of that, most of the previously reported epidemiological studies in humans have come up with inconsistent findings [2]. It is known that heterogeneity is a defining feature of TB, which is certainly common in molecular studies [43]. However, although the need for additional clinical evidence is obvious, disease phenotypes can possibly be determined by genotype features of specific strains, suggesting that different *M. tb* lineages could be more frequently present in specific clinical phenotypes and disease presentations than in others [2].

## Limitation

Spoligotyping has its limitations and may not truly detect ongoing changes (genetic differences) in a population and thereby not be the best tool for investigation of transmission networks [22]. Alternative molecular diagnostic tools, such as MIRU-VNTR and especially whole

genome sequencing, have shown to have better discriminatory power for investigating strain clustering and to confirm the ongoing rate of active TB disease transmission [14, 22]. Similarly, the fairly small sample size, uneven representation of strains from the study sites, and further categorization into different levels of factor variables, have reduced the power of our statistical analysis. Hence, the numerical estimates may not truly imitate the biological interaction or effect modification on host-related factors and specific *M. tb* lineages. Not only systematic and measurement errors, but the current study also recognized selection and recall bias where selected isolates were subjected for spoligotyping based molecular analysis. However, we have tried to minimize some of the anticipated measurement errors and known confounding effects. For instance, alongside with internal quality control procedures for the identification of lineages, SITVIT patterns were compared with alternative lineage classifications generated from linked databases (KBBN and CBN) and further verified using SNP based predictions. In addition, the multivariate analysis has considered and used to adjust the expected effect of regional variation on TB lineage predominance and related strain clustering.

## Conclusion and recommendation

Despite differences in geographical variations, the overall clustering suggested higher transmission of TB disease among human populations living around urban settings in Ethiopia. This Spoligotyping-based investigation showed that the rate of strain clustering was relatively higher among patients infected with L3 strains of *M. tb* as compared to L4. Regarding host-related factors, strain clustering rate was inversely associated with patients diagnosed with TB-HIV co-infection and comorbidity with other chronic illnesses. On the other hand, as compared to *M. tb* L4, active TB disease due to L3 strains was three times higher among TBLN patients and it was more likely to be associated with TB-HIV co-infection, while inversely associated with other concomitant chronic disease.

Altogether, the current findings add up to previous indications and contribute to evidence base on the continuous flux in the spectrum of TB infection and disease progression. Although it is difficult to be conclusive on a fixed categorical relationship between strain sub-lineages and disease type, as there is some other supportive evidence, disease phenotypes can possibly be determined by genotypic features of specific strains. Considering the complex pathogenesis of human TB disease and the interaction effect of other predisposing environmental factors, it seems that active infection due to specific *M. tb* lineages might be associated with specific clinical phenotypes and disease presentation.

Generally, considering the ongoing shift and heterogeneity of TB disease, clinical and public health interventions should be alongside with molecular evidence for targeting high-risk groups based on location, social determinants, disease comorbidities and related bacterial strain predominance. However, as the dynamics of socioeconomic transformations exert pressure on how people live and interact, large scale studies using advanced molecular techniques, like whole genome sequencing, should further reveal the degree to which the genetic variation influences disease epidemiology and phenotype in different population groups over time.

## Supporting information

**S1 Table. Spoligotype descriptions and lineage classifications of all clinical isolates included in this study.**
(XLS)

**S1 File.**
(DOCX)

**S1 Data.**
(DTA)

## Acknowledgments

We would like to forward our appreciation to supportive staff at the Armauer Hansen Research Institute and all members of the ETHICOBOTS project who had a great contribution to the success of this study. Besides, we would like to extend our acknowledgment to the University of Gondar and the academic staff of the public health institute. We also thank APHA for providing with membranes for spoligotyping.

The members of the Ethiopia Control of Bovine Tuberculosis Strategies (ETHICOBOTS) consortium are: Abraham Aseffa, Adane Mihret, Bamlak Tessema, Bizuneh Belachew, Eshcolewyene Fekadu, Fantanesh Melese, Gizachew Gemechu, Hawult Taye, Rea Tschopp, Shewit Haile, Sosina Ayalew, Tsegaye Hailu, all from Armauer Hansen Research Institute, Ethiopia; Rea Tschopp from Swiss Tropical and Public Health Institute, Switzerland; Adam Bekele, Chilot Yirga, Mulualem Ambaw, Tadele Mamo, Tesfaye Solomon, all from Ethiopian Institute of Agricultural Research, Ethiopia; Tilaye Teklewold from Amhara Regional Agricultural Research Institute, Ethiopia; Solomon Gebre, Getachew Gari, Mesfin Sahle, Abde Aliy, Abebe Olani, Asegedech Sirak, Gizat Almaw, Getnet Mekonnen, Mekdes Tamiru, Sintayehu Guta, all from National Animal Health Diagnostic and Investigation Centre, Ethiopia; James Wood, Andrew Conlan, Alan Clarke, all from Cambridge University, United Kingdom; Henrietta L. Moore and Catherine Hodge, both from University College London, United Kingdom; Constance Smith at University of Manchester, United Kingdom; R. Glyn Hewinson, Stefan Berg, Martin Vordermeier, Javier Nunez-Garcia, all from Animal and Plant Health Agency, United Kingdom; Gobena Ameni, Berecha Bayissa, Aboma Zewude, Adane Worku, Lemma Terfassa, Mahlet Chanyalew, Temesgen Mohammed, Yemisrach Zeleke, all from Addis ababa University, Ethiopia.

## Author Contributions

**Conceptualization:** Hawult Taye, Adane Mihret, James L. N. Wood, Stefan Berg, Abraham Aseffa.

**Data curation:** Kassahun Alemu, Adane Mihret, Sosina Ayalew, James L. N. Wood, Stefan Berg, Abraham Aseffa.

**Formal analysis:** Hawult Taye, Kassahun Alemu.

**Funding acquisition:** Adane Mihret, James L. N. Wood, Stefan Berg, Abraham Aseffa.

**Investigation:** Hawult Taye, Kassahun Alemu, Adane Mihret, Sosina Ayalew, Elena Hailu, Stefan Berg, Abraham Aseffa.

**Methodology:** Hawult Taye, Kassahun Alemu, Adane Mihret, Sosina Ayalew, Elena Hailu, James L. N. Wood, Ziv Shkedy, Stefan Berg, Abraham Aseffa.

**Project administration:** Hawult Taye, Adane Mihret, James L. N. Wood, Stefan Berg, Abraham Aseffa.

**Resources:** Adane Mihret, James L. N. Wood, Stefan Berg, Abraham Aseffa.

**Software:** Hawult Taye, Kassahun Alemu.

**Supervision:** Kassahun Alemu, Ziv Shkedy, Abraham Aseffa.

**Validation:** Hawult Taye, Kassahun Alemu, Adane Mihret, Sosina Ayalew, Elena Hailu, Stefan Berg, Abraham Aseffa.

**Visualization:** Hawult Taye, Kassahun Alemu, Adane Mihret, Sosina Ayalew, Abraham Aseffa.

**Writing – original draft:** Hawult Taye, Stefan Berg, Abraham Aseffa.

**Writing – review & editing:** Hawult Taye, Kassahun Alemu, Adane Mihret, Sosina Ayalew, James L. N. Wood, Ziv Shkedy, Stefan Berg, Abraham Aseffa.

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
