## [Decision Letter · Decision Letter 0]

3 Mar 2021

PONE-D-20-38732

Epidemiology of Mycobacterium tuberculosis and factors associated with strain clustering and lineage predominance around urban and peri-urban settings in Ethiopia.

PLOS ONE

Dear Dr. Adane,

Thank you for submitting your manuscript to PLOS ONE. After careful consideration, we feel that it has merit but does not fully meet PLOS ONE’s publication criteria as it currently stands. Therefore, we invite you to submit a revised version of the manuscript that addresses the points raised during the review process.

We look forward to receiving your revised manuscript.

Kind regards,

Md Jamal Uddin

Academic Editor

PLOS ONE

Journal Requirements:

3. One of the noted authors is a group or consortium [The ETHICOBOTS consortium]. In addition to naming the author group, please list the individual authors and affiliations within this group in the acknowledgments section of your manuscript. Please also indicate clearly a lead author for this group along with a contact email address.

**Please see attached pdf file to see the comments from editor.**

Reviewers' comments:

Reviewer's Responses to Questions

**Comments to the Author**

1. Is the manuscript technically sound, and do the data support the conclusions?

Reviewer #1: Yes

Reviewer #2: Partly

2. Has the statistical analysis been performed appropriately and rigorously? 

Reviewer #1: Yes

Reviewer #2: Yes

3. Have the authors made all data underlying the findings in their manuscript fully available?

Reviewer #1: Yes

Reviewer #2: Yes

4. Is the manuscript presented in an intelligible fashion and written in standard English?

Reviewer #1: Yes

Reviewer #2: Yes

5. Review Comments to the Author

**Reviewer #1: **The manuscript submitted by Taye et al. entitled "Epidemiology of Mycobacterium tuberculosis and factors associated with strain clustering and lineage predominance around urban and peri-urban settings in Ethiopia" is well written, technically sound and scientifically important to proper management of M. tuberculosis in Ethopia. But I have few minor comments, which should address before final acceptance of the manuscript.

1. At end of the title there is a punctuation (.) which have to delete

2. In the abstract authors time to time used abbreviation (PTB, TBLN, SITVIT2, SIT) which is difficult to understand by the general reader. Thus, an elaboration should include.

3. In the Methods of the abstract authors mentioned that they have collected the sample from four different regions in Ethiopia were recruited in the year 2016 and 2017. But why they have publishing this result after 3 years. Need an explanation for that because within this time period lineage of microorganism may change. In addition, it is prescribed to mention the exclusion and inclusion criteria of TB patients recruitment.

4. From the title its clear that authors wanted to find out epidemiology and factors associated with strain clustering and lineage predominance but in the introduction they failed to good use of literature review. They should clearly include what are the possible factors that can be associate and with brief introduction of underlying mechanism. For example, they found that L3 M. tuberculosis strains were more likely to be associated with TBLN and TB-HIV co-infection, but what is the possible mechanism of this association and co-infection. Why M. tuberculosis (bacteria) have association with HIV (virus).

5. For ethical consideration, need to add reference no. of the ethical clearance

6. In the result section, if (optional) the sequencing data of all strains in different geographic region of Ethiopia is available, a phylogenetic tree can represent the closeness of the strains and also better understanding of lineage.

6. In the discussion, authors need elaborate on what are the underlying factors that contributed to prevail more TB in the urban than rural area.

7. In the discussion (Line 275-276), author mentioned unique pattern considered to have resulted from reactivation of latent infection. But what about adaptation of those strains with environment and genetical changes (mutation/polymorphisms)?

**Reviewer #2: **It is my pleasure to review such a well written manuscript. The data were rigorously analyzed. It involved a great deal of work and dedication to complete this work and it is clearly visible. However, I would like to ask one question to the authors and that would be on sample size calculation. I could not find any clear description on how the investigators arrived at the sample size used in this analysis. The purposive selection of sites coupled with lack of information on sample size calculation makes it difficult to accept the results of such rigorous analysis and also puts the question of generalizability forward. Are these analyses generalizable to the whole community or the regions? We do not know and this information is vital to the analysis. I am sure the authors will be able to answer the question. If it is an exploratory study and the sample size calculation was not done rigorously. I would request the authors to include this information in the limitations section explicitly to make the readers aware of the fact. I congratulate the authors for their hard work and look forward to see their response. Thanks!

6. PLOS authors have the option to publish the peer review history of their article (what does this mean?). If published, this will include your full peer review and any attached files.

Reviewer #1: **Yes: **Prof. Dr. Mohammad Jakir Hosen

Reviewer #2: **Yes: **SHAHRIAR AHMED

---

## [Decision Letter · Decision Letter 1]

21 May 2021

PONE-D-20-38732R1

Epidemiology of Mycobacterium tuberculosis lineages and strain clustering within urban and peri-urban settings in Ethiopia

PLOS ONE

Dear Dr. Adane,

Thank you for submitting your manuscript to PLOS ONE. After careful consideration, we feel that it has merit but does not fully meet PLOS ONE’s publication criteria as it currently stands. Therefore, we invite you to submit a revised version of the manuscript that addresses the points raised during the review process.

As requested by academic editor, in online supplements, the author did not provide all necessary software codes, including detailed model fitting criteria, particularly for hierarchical (multi-level) logistic regression. In this case, the author must provide such information, and if such information is not provided, the manuscript will not be considered for further steps.

Please keep the conclusion and recommendations separate. First, provide possible recommendations including clinical implications based on your key findings, and then provide conclusions, which are your article's take-home message.

In the abstract, AOR and CI needs to elaborate first.

Please avoid using older references and instead try to use more recent and relevant ones.

We look forward to receiving your revised manuscript.

Kind regards,

Md Jamal Uddin

Academic Editor

PLOS ONE

Journal Requirements:

Additional Editor Comments (if provided):

Thank you for improving the manuscript as requested by reviewers and editor. Still several things need to improve.

As requested by academic editor, in online supplements, the author did not provide all necessary software codes, including detailed model fitting criteria, particularly for hierarchical (multi-level) logistic regression. In this case, the author must provide such information, and if such information is not provided, the manuscript will not be considered for further steps.

Please keep the conclusion and recommendations separate. First, provide possible recommendations including clinical implications based on your key findings, and then provide conclusions, which are your article's take-home message.

In the abstract, AOR and CI needs to elaborate first.

Please avoid using older references and instead try to use more recent and relevant ones.

Reviewers' comments:

Reviewer's Responses to Questions

**Comments to the Author**

1. If the authors have adequately addressed your comments raised in a previous round of review and you feel that this manuscript is now acceptable for publication, you may indicate that here to bypass the “Comments to the Author” section, enter your conflict of interest statement in the “Confidential to Editor” section, and submit your "Accept" recommendation.

Reviewer #1: All comments have been addressed

Reviewer #2: All comments have been addressed

2. Is the manuscript technically sound, and do the data support the conclusions?

Reviewer #1: Yes

Reviewer #2: Yes

3. Has the statistical analysis been performed appropriately and rigorously? 

Reviewer #1: Yes

Reviewer #2: Yes

4. Have the authors made all data underlying the findings in their manuscript fully available?

Reviewer #1: Yes

Reviewer #2: Yes

5. Is the manuscript presented in an intelligible fashion and written in standard English?

Reviewer #1: Yes

Reviewer #2: Yes

6. Review Comments to the Author

Reviewer #1: Current form of manuscript is better than before, novel work, new findings, research and publication ethics strictly maintained, strongly recommended for acceptance.

Reviewer #2: Thank you for addressing the comments. I am satisfied with the responses and the revisions. This is an important piece of work and I feel proud to be associated with this. Best of luck!

7. PLOS authors have the option to publish the peer review history of their article (what does this mean?). If published, this will include your full peer review and any attached files.

Reviewer #1: **Yes: **Prof. Dr. Mohammad Jakir Hosen

Reviewer #2: **Yes: **SHAHRIAR AHMED

---

## [Author Response · Author response to Decision Letter 1]

25 May 2021

Dear professor Md Jamal Uddin,

We thank the valuable feedback and comments from the two reviewers and academic editors that help us to improve the first version of the manuscript (PONE-D-20-38732R). We are happy that two of the reviewers acknowledged as all first round comments have been addressed. 

Recently we received additional comments from the academic editor for the revised manuscript (PONE-D-20-38732R1) entitled “Epidemiology of Mycobacterium tuberculosis lineages and strain clustering within urban and peri-urban settings in Ethiopia”.

As requested by academic editor, all second round comments and remarks were addressed and changes in the manuscript were done accordingly. We made the required revision and both a 'clean' version (Revised manuscript) and the manuscript with all changes tracked are uploaded and submitted. 

In the recent version, we replaced older references and retracted articles which were mistakenly included in former version. Here the ‘software codes’ and 'Response to Reviewers' are also uploaded. 

We thank you for considering the revised manuscript for publication in your Journal.

Kind regards 

Hawult Taye Adane 

Email: hawultachew@gmail.com

---

## [Editor Report · Decision Letter 2]

7 Jun 2021

Epidemiology of Mycobacterium tuberculosis lineages and strain clustering within urban and peri-urban settings in Ethiopia

PONE-D-20-38732R2

Dear Dr. Adane,

We’re pleased to inform you that your manuscript has been judged scientifically suitable for publication and will be formally accepted for publication once it meets all outstanding technical requirements.

Kind regards,

Md Jamal Uddin

Academic Editor

PLOS ONE
---

## [Editor Report · Acceptance letter]

2 Jul 2021

PONE-D-20-38732R2 

Epidemiology of *Mycobacterium tuberculosis* lineages and strain clustering within urban and peri-urban settings in Ethiopia 

Dear Dr. Taye:

I'm pleased to inform you that your manuscript has been deemed suitable for publication in PLOS ONE. Congratulations! Your manuscript is now with our production department. 

Kind regards, 

on behalf of

Dr. Md Jamal Uddin 

Academic Editor

PLOS ONE